# Inverse Identification of a Constitutive Model for High-Speed Forming Simulation: An Application to Electromagnetic Metal Forming

**DOI:** 10.3390/ma15207179

**Published:** 2022-10-14

**Authors:** Dayoung Kang, Hak-Gon Noh, Jeong Kim, Kyunghoon Lee

**Affiliations:** 1Department of Aerospace Engineering, Pusan National University, Busan 46241, Korea; 2Korea Aerospace Industries, Sacheon 52537, Korea

**Keywords:** inverse identification, high-speed forming, electromagnetic metal forming, regularized nonlinear least squares, model order reduction

## Abstract

Forming simulation requires a constitutive model whose parameters are typically determined with tensile tests assumed static. However, this conventional approach is impractical for high-speed forming simulation characterized by high strain rates inducing transient effects. To identify constitutive parameters in relation to high-speed forming simulation, we formulated the problem of constitutive modeling as inverse parameter estimation addressed by regularized nonlinear least squares. Regarding the proposed inverse constitutive modeling, we adopted the *L*-curve method for proper regularization and model order reduction for rapid simulation. For demonstration, we corroborated the proposed strategy by identifying the modified Johnson–Cook model in the context of a free bulge test with electromagnetic metal forming simulation. The *L*-curve method allowed us to systematically choose a regularization parameter, and model order reduction brought enormous computational savings. After identifying constitutive parameters, we successfully verified and validated the reduced and original simulation models, respectively, with a manufactured workpiece. In addition, we validated the numerically identified constitutive model with a dynamic material test using a split Hopkinson pressure bar. Overall, we showed that inverse constitutive modeling for high-speed forming simulation can be effectively tackled by regularized nonlinear least squares with the help of an *L*-curve and a reduced-order model.

## 1. Introduction

A constitutive model, also known as a material model, is indispensable for simulating a forming process, as it predicts the plastic behavior of a material in relation to forming factors, such as strain, temperature, and strain rate. A constitutive model normally hinges on constitutive parameters, also referred to as material constants, that need to be determined for a material of interest. In general, if a forming process involves strain rates less than 1/s, constitutive parameters can be found with flow stress measurements attained by quasi-static tensile tests. However, if a forming process entails strain rates higher than 1/s, dynamic effects start to kick in so that material properties vary drastically due to material inertia and stress wave transmission. As a consequence, conventional constitutive modeling with tensile tests is inadequate for high-speed forming technology, such as electromagnetic metal forming (EMF) and electrohydraulic forming (EHF) [1,2].

To construct a constitutive model in conjunction with a high-speed forming simulation, many researchers have strived with experimental as well as numerical endeavors. To begin with experimental approaches, disparate experimental techniques are known to be suitable for different ranges of strain rates. According to Ref. [3], valid strain rate ranges per experimental instruments can be summarized as follows: (i) the split Hopkinson pressure bar (SHPB) test [4,5]: 1 × 10^2^/s–1 × 10^4^/s, (ii) the Taylor test [5,6]: 1 × 10^4^/s–1 × 10^6^/s, and (iii) the inverse flyer plate test: 1 × 10^6^/s–1 × 10^9^/s. In previous literature, an SHPB was applied to steel and aluminum alloys at strain rates ranging from 180/s–500/s and from 700/s–1500/s, respectively [7]. Although experimental techniques allow one to directly measure stress-strain curves at high strain rates, they tend to demand expensive, bulky experimental apparatuses.

In contrast, numerical approaches can dispense with measurement equipment because they only necessitate a simulation along with a manufactured workpiece. For constitutive modeling, numerical techniques attempt to indirectly identify constitutive parameters that enable a simulation to anticipate a deformed shape congruent with the actual workpiece; after all, computational approaches for constitutive modeling boil down to an inverse problem [8,9]. In the literature, the Cowper–Symonds parameters were determined with the Taylor impact test by nonlinear least squares [3]. Similarly, the parameters of various constitutive models, such as the Johnson–Cook and Zerilli–Armstrong models, were estimated with the SHPB test in a least mean square sense [10,11]. Recently, the Cowper–Symonds parameters were also identified with a free bulge test using EMF via nonlinear least squares [12] and using EHF via Bayesian inference [13]. Deterministic inverse formulations, such as least squares estimation, may not properly address the ambiguity of parameter estimates persistent in an inverse problem. On the other hand, non-deterministic inverse formulations, such as Bayesian inverse estimation, may handle uncertainties in parameter estimates, but they need an enormous number of simulations.

In pursuit of numerical studies, we propose a strategy of inverse constitutive modeling for high-speed forming simulation in the form of nonlinear least squares with regularization. We introduce regularization to nonlinear least squares as a vehicle to alleviate the ill-posedness of inverse estimation in finding constitutive parameters. To facilitate regularization parameter selection, we capitalize on the *L*-curve method, which trades off the changes of residual and solution norms with respect to the variation of a regularization parameter. Since the *L*-curve technique is heuristic, such that it requires a series of solutions of nonlinear least squares problems at different regularization parameters, computational cost arises as a primary concern. Therefore, to surmount this issue, one may rely on advanced finite element methods (FEMs) adopted for efficient EMF simulation [14,15,16]. Instead, we utilize model order reduction (MOR) [17,18], which exploits a low-dimensional structure of a large-dimensional simulation output for expeditive evaluation.

For demonstration, we apply the proposed scheme to the identification of the modified Johnson–Cook model in the context of a free bulge test using electromagnetic force, i.e., Lorentz force. As a follow-up study, this research is similar to the preliminary one [12] in that inverse constitutive modeling is addressed for the EMF simulation of a free bulge test with a rapid simulation built upon MOR. On the other hand, this research is dissimilar to the previous one in the material, in the constitutive model, and in the inverse estimation formulation. More importantly, we present comprehensive mathematical formulations not restricted to EMF but applicable to high-speed forming technology in general.

Overall, this paper is structured as follows. After the introduction, Section 2 reviews the theoretical background on inverse parameter estimation as well as model approximation by MOR with two statistical techniques: principal component analysis (PCA) and Kriging [19]. Afterward, Section 3 delineates the experimental and numerical realization of a free bulge test with EMF, followed by Section 4 on the construction and verification of an approximate yet rapid EMF simulation. Next, Section 5 illustrates the proposed tactic for inverse constitutive modeling with computational and experimental models of the free bulge test and subsequently verifies the reduced EMF simulation, then validates the original EMF simulation. Section 6 compares a flow stress curve predicted numerically by the identified constitutive model to that acquired experimentally by a dynamic material test with an SHPB. Lastly, Section 7 summarizes this paper and finishes with conclusions, contributions, and future work.

## 2. Formulations

In this section, we formulate an inverse problem for constitutive parameter estimation with a forming simulation and then recast the same problem with a rapid approximation of the original EMF simulation. Inverse parameter estimation was addressed in the form of regularized nonlinear least squares that compromises residual and regularization terms from a deterministic perspective. For model approximation, we adopted MOR relying on PCA and Kriging. The former is to identify a basis capturing a low-dimensional structure of simulation output, and the latter is to estimate basis coefficients accordingly for parametric variation.

### 2.1. Inverse Identification

#### 2.1.1. Forward Model

Consider a forming simulation $f:R\to R^{m}$ expressed as: (1)y=f(x;σ),
where $y\in R^{m}$ is a simulation output, such as displacements, $x\in R^{d}$ are the discretized coordinates of a computational domain, and σ∈R is flow stress. In Equation (1), flow stress σ is achieved by a constitutive model $g:R^{p}\to R$ defined by: (2)σ=g(w;θ),
where $w\in R^{k}$ are forming factors affecting flow stress, and $\theta \in R^{p}$ are constitutive parameters that need to be inferred. For instance, a constitutive model may depend on a temperature T∈R, a strain ε∈R, and a strain rate $\dot{\varepsilon }\in R$; hence, $w=\left[T,\varepsilon ,\dot{\varepsilon }\right]\in R^{3}$. Typically, constitutive parameters θ may vary with diverse constitutive models and different materials.

By the composition of *f* and *g* in Equations (1) and (2), respectively, we can define a forward model $h:R^{p}\to R^{m}$ given *x* and *w* such that: (3)y=(f∘g)(θ)=h(θ),
which directly associates *y* with θ through *h*. Since a forming simulation *f* is computationally expensive in general, we draw on an approximate model $\tilde{f}:R\to R^{m}$ in lieu of *f* for rapid evaluation assuming an additive error $e:R\to R^{m}$ as follows: (4)$$f\left(x;\sigma \right)=\tilde{f}\left(x;\sigma \right)+e_{f}\left(x;\sigma \right).$$

By substituting *f* in Equation (4) into *h* in Equation (3), we obtain: (5)$$y=\left(f\circ g\right)\left(\theta \right)=\left(\left(\tilde{f}+e_{f}\right)\circ g\right)\left(\theta \right)=\left(\tilde{f}\circ g\right)\left(\theta \right)+\left(e_{f}\circ g\right)\left(\theta \right)=\tilde{h}\left(\theta \right)+e_{h}\left(\theta \right),$$
which results in an approximate forward model $\tilde{h}:R^{p}\to R^{m}$ along with an approximate-model-induced error $e_{h}:R^{p}\to R^{m}$.

#### 2.1.2. Regularized Nonlinear Least Squares

Consider the measured configuration of a manufactured workpiece $y_{o}\in R^{m}$, corresponding to a predicted configuration $y\in R^{m}$ resulting from a forward model *h* in Equation (3). From an inverse problem viewpoint, we are interested in indirectly determining constitutive parameters θ that cause *h* to produce *y* as close as $y_{o}$. To address this estimation problem, we formulate a problem of nonlinear least squares with regularization such that: (6)$$\min_{\theta }∥y_{o}{-h\left(\theta \right)∥}^{2}+\alpha^{2}{∥\theta ∥}^{2},$$
which is the weighted sum of a squared residual norm $∥y_{o}{-h\left(\theta \right)∥}^{2}$ and a squared solution norm ${∥\theta ∥}^{2}$. In Equation (6), α∈R is a regularization parameter that controls the magnitude of the second term to impose a penalty preventing θ from growing. In detail, the former is a residual term evaluating the misfit between $y_{o}$ and *y*, whereas the latter is a regularization term enforcing Occam’s razor on finding θ. The regularization term α is appended as a remedy to ease the inherent ill-posedness of inverse estimation. For instance, the increase of α tends to smooth out the first misfit term in Equation (6), which is conducive to locating θ. However, if α is too large, it may distort the shape of the misfit term, which causes the regularized problem to deviate from the original problem, leading to irrelevant θ.

Since the regularized nonlinear least squares in Equation (6) necessitates nonlinear optimization, it is inevitable to repeatedly invoke a forward model *h* that is usually time-consuming. Therefore, to effectively deal with Equation (6), we adopt an approximate forward model $\tilde{h}$ instead of the original forward model *h*. For this purpose, we may utilize surrogate modeling or MOR techniques. Assuming $e_{h}$ in Equation (5) due to the use of $\tilde{f}$ is negligible, we can approximate the residual term in Equation (6) as $y_{o}-h\left(\theta \right)=y_{o}-\tilde{h}\left(\theta \right)-e_{h}\left(\theta \right)\approx y_{o}-\tilde{h}\left(\theta \right).$ As a result, the regularized nonlinear least squares problem in Equation (6) is rephrased to: (7)$$\min_{\theta }∥y_{o}-\tilde{h}{\left(\theta \right)∥}^{2}+\alpha^{2}{∥\theta ∥}^{2}.$$

Regarding the use of $\tilde{h}$ in Equation (7), one caveat that should be mentioned is that the solution of Equation (7) is not guaranteed to be the same as that of Equation (6), which is typical when a surrogate is employed in seeking efficiency at the expense of accuracy. To wit, Equation (7) is technically inconsistent with Equation (6) unless $e_{h}$ is independent of θ. Consequently, accuracy loss in the solution is inescapable. Thus, constructing $\tilde{h}$ as close to *h* as possible is imperative to mitigate the aftereffect of using $\tilde{h}$. It is also necessary to verify the agreement of $\tilde{h}$ with *h* at the solution.

### 2.2. Model Order Reduction

#### 2.2.1. Low-Dimensional Approximation

Model order reduction (MOR) [17] is an approximation technique that is particularly useful to expedite a simulation whose output is large dimensional, e.g., at least $O(10^{2})$ or $O(10^{3})$. This numerical technique benefits from a low dimensional structure of large dimensional data over a confined domain of inputs. Here, we briefly delineate the formulation of specifically hinging on PCA for the low dimensional structure.

Suppose that we are interested in the variation of a simulation output $y\left(x;\theta \right)\in R^{m}$ with respect to inputs $\theta \in R^{p}$ within a domain of interest $D=\left[\theta_{l}\times \theta_{u}\right]\subset R^{p}$, where $\theta_{l}\in R^{p}$ and $\theta_{u}\in R^{p}$ are the lower and upper bounds of θ, respectively. In our application, *y* and θ are forming simulation output and constitutive parameters in Equation (3). Let $y_{c}\left(x;\theta \right)\in R^{m}$ be a mean-subtracted simulation output defined by $y_{c}\left(x;\theta \right)=y\left(x;\theta \right)-\mu_{y}\left(x\right)$, where $\mu_{y}\left(x\right)\in R^{m}$ is the mean of *y* over D. Typically, $\mu_{y}$ is estimated by a sample mean $\bar{y}\left(x\right)=\frac{1}{n}\sum_{i=1}^{n}y\left(x;\theta_{i}\right)\in R^{m}$, where $\theta_{i}\in S_{\theta }={\left\{\theta_{i}\right\}}_{i=1}^{n}\subset D$. Henceforth, we treat *y* as $y_{c}$ for the sake of convenience. In view of MOR, we presume that *y* is a vector in a vector space spanned by a basis ${\left\{u_{i}\left(x\right)\right\}}_{i=1}^{r}\in R^{m\times r}$ for θ∈D. Provided that *y* admits the separation of *x* and θ, we can express *y* as a linear combination of $u_{i}\left(x\right)\in R^{m}$ such that: (8)$$y\left(x;\theta \right)=\sum_{i=1}^{r}a_{i}\left(\theta \right)u_{i}\left(x\right),\;\forall \theta \in S_{\theta }\subset D,$$
where $a_{i}\left(\theta \right)\in R$ is the *i*th basis (or weighting) coefficient, which can be easily evaluated by the orthogonal projection of y(x;θ) onto a vector space spanned by $u_{i}\left(x\right)$. Note that significant dimensionality reduction can be achieved from *m* to *r*, for normally r≪m.

For the effective representation of *y*, we may use the first leading *q* basis vectors after ignoring trifling r−q basis vectors. Namely, we approximate *y* on a low dimensional structure captured by ${\left\{u_{i}\left(x\right)\right\}}_{i=1}^{q}\in R^{m\times q}$, a subspace of the vector space spanned by ${\left\{u_{i}\left(x\right)\right\}}_{i=1}^{r}$, as follows: (9)$$y\left(x;\theta \right)\approx \sum_{i=1}^{q}a_{i}\left(\theta \right)u_{i}\left(x\right),\;\forall \theta \in S_{\theta }\subset D.$$

To predict *y* at an unseen parameter $\theta_{0}\in D\backslash S_{\theta }$, we assume that $y(x;\theta_{0})$ is also spanned by ${\left\{u_{i}\left(x\right)\right\}}_{i=1}^{r}$ and is approximated with the *q* leading basis vectors as before, such that: (10)$$y\left(x;\theta_{0}\right)\approx \sum_{i=1}^{q}a_{i}\left(\theta_{0}\right)u_{i}\left(x\right),\;\forall \theta_{0}\in D\backslash S_{\theta },$$
where $a_{i}\left(\theta_{0}\right)$ needs to be estimated unlike before. In general, a basis ${\left\{u_{i}\left(x\right)\right\}}_{i=1}^{r}$ is unavailable a priori, thus we leverage PCA to empirically achieve an orthonormal basis with a set of simulation outputs ${\left\{y\left(x;\theta_{i}\right)\right\}}_{i=1}^{n}$ obtained with $\theta_{i}\in S_{\theta }={\left\{\theta_{i}\right\}}_{i=1}^{n}$. Likewise, a basis coefficient $a_{i}$ is unknown for $\theta_{0}\in D\backslash S_{\theta }$ either, so we utilize Kriging to empirically construct a functional relationship between $a_{i}$ and $\theta_{0}$.

#### 2.2.2. Principal Component Analysis

The first component of MOR adopted in this research is PCA for basis extraction. From a statistics perspective, PCA formulates the concept of change of basis to effectively describe the given data as follows. Suppose that a simulation output $y_{i}=y\left(x;\theta_{i}\right)\in R^{m}$ is a realization of a random vector. Let *Y* be an output collection such that $Y=\left[y_{1},\ldots ,y_{n}\right]\in R^{m\times n}$. As before, in Section 2.2.1, we treat $y_{i}$ and *Y* as mean-shifted quantities evaluated by $y_{i}-\bar{y}$ and $Y-\bar{y}1\;\;1_{n}^{T}$, respectively, where $\bar{y}\in R^{m}$ is a sample mean defined by $\bar{y}=\frac{1}{n}\sum_{i=1}^{n}y_{i}\in R^{m}$, and $1\;\;1_{n}$ is a vector of *n* ones given by $1\;\;1_{n}=\left[1;\ldots ;1\right]\in R^{n}$.

To succinctly represent $y_{i}$ by means of a basis change, we aim to find a different basis vector $u_{k}\in R^{m}$. The coordinate of $y_{i}$ in the direction of $u_{k}$ can be obtained by the orthogonal projection of $y_{i}$ onto $u_{k}$ such that $a_{ki}=\left(y_{i},u_{k}\right)=u_{k}^{T}y_{i}\in R$; here, $a_{ki}$ is referred to as the *k*th principal component of $y_{i}$, and $u_{k}$ is called the *k*th principal direction. Let $a_{k}$ be a vector composed of the *k*th principal components such that $a_{k}=\left[u_{k}^{T}y_{1};\ldots ;u_{k}^{T}y_{n}\right]=Y^{T}u_{k}\in R^{n}$. Then, the sample variance of $a_{k}$ is evaluated by: (11)$$\begin{array}{cc} s_{a_{k}}^{2} & =\frac{1}{n}\sum_{i=1}^{n}{\left(a_{ki}\right)}^{2}=\frac{1}{n}\sum_{i=1}^{n}{\left(u_{k}^{T}y_{i}\right)}^{2}=\frac{1}{n}\sum_{i=1}^{n}\left(u_{k}^{T}y_{i}\right){\left(u_{k}^{T}y_{i}\right)}^{T}\\ \; & =u_{k}^{T}\left(\frac{1}{n}\sum_{i=1}^{n}y_{i}y_{i}^{T}\right)u_{k}=u_{k}^{T}\left(\frac{1}{n}YY^{T}\right)u_{k}=u_{k}^{T}Su_{k}\in R, \end{array}$$
where $S\in R^{m\times m}$ is the sample covariance matrix of *Y*.

Since we are interested in effectively representing $y_{i}$ with respect to $u_{k}$, we need to find $u_{k}$ that can account for the sample variance $s_{a_{k}}^{2}$ as much as possible. In other words, we are required to solve the following optimization problem: (12)$$\max_{u_{k}}u_{k}^{T}Su_{k}\;\mathrm{subject}\;\mathrm{to}\;\left(u_{j},u_{k}\right)=\delta_{jk},$$
where j=1,…,k. In Equation (12), the constraint $\left(u_{j},u_{k}\right)=\delta_{jk}$ is enforced for principal directions being orthonormal. This constrained optimization problem in Equation (12) can be recast into an unconstrained optimization problem with the help of Lagrange multipliers. The Lagrangian associated with Equation (12) is formed as: (13)$$L\left(u_{k},\lambda_{1},\ldots ,\lambda_{k}\right)=u_{k}^{T}Su_{k}-\sum_{j=1}^{k}\lambda_{j}\left[\left(u_{j},u_{k}\right)-\delta_{jk}\right],$$
and the first derivative of L with respect to $u_{k}$ is required to vanish for optimal $u_{k}$   being derived.
(14)$$\frac{\partial L}{\partial u_{k}}=2Su_{k}-2\lambda_{k}u_{k}-\sum_{j=1}^{k-1}\lambda_{j}u_{j}=0,$$
which results in,
(15)$$Su_{k}=\lambda_{k}u_{k}+\frac{1}{2}\sum_{j=1}^{k-1}\lambda_{j}u_{j}.$$

To evaluate $\lambda_{j}$, we take $u_{j}^{T}$ to both sides of Equation (14) as below: (16)$$2u_{j}^{T}Su_{k}-2\lambda_{k}u_{j}^{T}u_{k}-\lambda_{j}u_{j}^{T}u_{j}=2u_{j}^{T}Su_{k}-\lambda_{j}u_{j}^{T}u_{j}=0,$$
which gives $\lambda_{j}=0$ because of $u_{j}^{T}S=\lambda_{j}u_{j}^{T}$ from $Su_{j}=\lambda_{j}u_{j}$ for j<k. Thus, we obtain ${\left\{\lambda_{j}\right\}}_{j=1}^{k-1}=0$. Finally, the optimization problem in Equation (12) reduces to: (17)$$Su_{k}=\lambda_{k}u_{k},$$
which is an eigenvalue problem to find an eigenvalue $\lambda_{k}$ and a corresponding eigenvector $u_{k}$.

As with a principal direction $u_{k}$ related to the column vectors of *Y*, we may find a principal direction $v_{k}\in R^{n}$ associated with the column vectors of $Y^{T}$, i.e., the row vectors of *Y*. Similar to Equation (12), a principal direction $v_{k}\in R^{n}$ that maximizes the sample variance of the *k*th principal components of $Y^{T}$ can be found by: (18)$$\max_{v_{k}}v_{k}^{T}Rv_{k}\;\mathrm{subject}\;\mathrm{to}\;\left(v_{j},v_{k}\right)=\delta_{jk}\;\mathrm{for}\;\;j=1,\ldots ,k,$$
where $R\in R^{n\times n}$ is the sample covariance matrix of $Y^{T}$. As with Equation (17), the optimization problem in Equation (18) turns into: (19)$$Rv_{k}=\lambda_{k}v_{k},$$
and we can restore a principal direction $u_{k}$ from $v_{k}$ via the singular value decomposition of *Y* as follows: (20)$$u_{k}=\frac{1}{\sqrt{\lambda_{k}}}Yv_{k}.$$

Note that Equation (19) is preferred if m≫n, and so is Equation (17) if m≪n for computational efficiency associated with sample covariance evaluation. In the context of our application, we used the former since m=120 and n=20.

#### 2.2.3. Kriging Interpolation

The second component of MOR used for this research is Kriging for basis coefficient evaluation. We adopted Kriging interpolation as a means of function approximation to estimate each basis coefficient at unseen constitutive parameters. Kriging is a statistical technique that originated in geostatistics for mining and has been widely disseminated to other fields for its superior interpolation capability.

From a statistical standpoint, Kriging views an observation y(x)∈R at an input $x\in R^{p}$ as a realization of a Gaussian random variable Y(x) expressed by: (21)Y(x)=f(x)+Z(x),
where f(x) and Z(x) are deterministic and stochastic parts, respectively. In our application, a scalar output *y* dependent on an input *x* corresponds to the *i*th basis coefficient $a_{i}$ relying on constitutive parameters θ. Kriging formulations are categorized by different assumptions on the deterministic part f(x), and here we employed *ordinary* Kriging, which presumes that f(x) is unknown and constant such that f(x)=μ∈R.

According to the Kriging formulation, a set of observations ${\left\{y\left(x_{i}\right)\right\}}_{i=1}^{n}$ stems from a multivariate random variable ${\left\{Y\left(x_{i}\right)\right\}}_{i=1}^{n}$. To formulate the probability density function of ${\left\{Y\left(x_{i}\right)\right\}}_{i=1}^{n}$, we presume that ${\left\{Z\left(x_{i}\right)\right\}}_{i=1}^{n}$ is distributed as a zero-mean Gaussian random variable $N({}_{n},\sigma^{2}\Psi )$, where ${}_{n}\in R^{n}$ is a vector of *n* zeros, $\sigma^{2}\in R$ is a process variance, and $\Psi \in R^{n\times n}$ is a correlation matrix. Note that Ψ hinges on a correlation function $\psi (x_{i},x_{j};\theta )$, where $x_{i}$ and $x_{j}$ are the *i*th and *j*th inputs, respectively, and $\theta \in R^{p}$ is a hyperparameter that determines correlation strength based on the relative distance between $x_{i}$ and $x_{j}$. Since the stochastic nature of ${\left\{Y\left(x_{i}\right)\right\}}_{i=1}^{n}$ results from that of ${\left\{Z\left(x_{i}\right)\right\}}_{i=1}^{n}$, ${\left\{Y\left(x_{i}\right)\right\}}_{i=1}^{n}$ is also found as a Gaussian random variable following $N(\mu 1\;\;1_{n},\sigma^{2}\Psi )$, where $1\;\;1_{n}\in R^{n}$ is a vector of *n* ones.

After formulating the probability distribution of ${\left\{Y\left(x_{i}\right)\right\}}_{i=1}^{n}$, we construct a likelihood function: (22)$$L\left(\mu ,\sigma^{2},\theta |y\right)=\frac{1}{{\left(2\pi \sigma^{2}\right)}^{n/2}{\left|\Psi \right|}^{1/2}}\exp\left(-\frac{{\left(y-\mu 1\;\;1_{n}\right)}^{T}\Psi^{-1}\left(y-\mu 1\;\;1_{n}\right)}{2\sigma^{2}}\right),$$
where $y=\left[y\left(x_{1}\right);\ldots ;y\left(x_{n}\right)\right]\in R^{n}$, to estimate parameters, such as μ, $\sigma^{2}$, and θ. By applying the method of maximum likelihood (ML) to the log-transformed likelihood function lnL, we obtain the ML estimates of μ and $\sigma^{2}$ as shown below: (23)$$\hat{\mu }=\frac{1\;\;1^{T}\Psi^{-1}\;y}{1\;\;1_{n}^{T}\Psi^{-1}1\;\;1_{n}},\;\mathrm{and}\;\hat{\sigma }{}^{2}=\frac{{\left(y-\hat{\mu }1\;\;1_{n}\right)}^{T}\Psi^{-1}\left(y-\hat{\mu }1\;\;1_{n}\right)}{n}.$$

Unlike the probability parameters μ and $\sigma^{2}$, the hyperparameter θ cannot be derived analytically by the ML method because lnL is nonlinear in θ. Therefore, to obtain the ML estimate of θ, we address the nonlinear optimization of a concentrated log-likelihood function: (24)$$\ln L_{c}\left(\theta |y\right)\approx -\frac{n}{2}\ln{\hat{\sigma }}^{2}-\frac{1}{2}\ln\left|\Psi \right|,$$
which is formed after the substitution of $\hat{\mu }$ and ${\hat{\sigma }}^{2}$ into lnL.

Finally, to predict $y(x_{0})$ at an unobserved input $x_{0}\notin {\left\{x_{i}\right\}}_{i=1}^{n}$, we presume that $\tilde{y}=\left[y;y\left(x_{0}\right)\right]\in R^{n+1}$, i.e., *y* augmented with $y(x_{0})$, shares the same Gaussian distribution with *y*. Thus, $\tilde{y}$ is distributed as $N(\hat{\mu }1\;\;1_{n+1},{\hat{\sigma }}^{2}\tilde{\Psi })$, where $\tilde{\Psi }\in R^{\left(n+1)\times (n+1\right)}$ is an augmented correlation matrix evaluated with $\left\{x_{1},\ldots ,x_{n},x_{0}\right\}\in R^{n+1}$. As with the ML parameter estimation, an estimate of $y(x_{0})$ can be achieved by the method of ML as follows: $$\hat{y}\left(x_{0}\right)=\hat{\mu }+\psi^{T}\Psi^{-1}\left(y-\hat{\mu }1\;\;1_{n}\right),$$
where $\psi ={\left[\psi \left(x_{1},x_{0}\right),\ldots ,\psi \left(x_{n},x_{0}\right)\right]}^{T}\in R^{n}$. Note that a mean-subtracted prediction estimate $\hat{y}\left(x_{0}\right)-\hat{\mu }\in R$ is composed of a basis $\Psi^{-1}\left(y-\hat{\mu }1\;\;1_{n}\right)\in R^{n}$ and basis coefficients $\psi \in R^{n}$; the former is generated with training inputs ${\left\{x_{i}\right\}}_{i=1}^{n}$, whereas the latter is produced with the known training inputs as well as an unseen input $x_{0}$.

## 3. Free Bulge Test with Electromagnetic Force

We are concerned with experimentally and numerically setting up a free bulge test with EMF that was chosen as an epitome of high-speed forming technology. We first delineate the experimental procedure along with the apparatus and then describe the coupled structural-electromagnetic modeling and analysis. The EMF simulation of a free bulge test was implemented by LS-DYNA with the electromagnetic (EM) module and dependent on the modified Johnson–Cook model.

### 3.1. Electromagnetic Forming Experiment

#### 3.1.1. Experimental Procedure

For a free bulge test, we utilized an in-house built EMF apparatus, named PNU-32, which consists of the three pieces of equipment as depicted in Figure 1: (i) a forming die to deform a workpiece by electromagnetic force; (ii) a control system to regulate input current strength; (iii) and a capacitor bank consisting of 30 capacitors to store electric energy. As schematically illustrated in Figure 2, the EMF apparatus Figure 1 can be represented as an resistance, inductance, and capacitance (RLC) equivalent circuit assembled by several electrical elements: equivalent resistance *R* and equivalent inductance *L* associated with the entire EMF apparatus, capacitance *C* linked with the capacitor assembly, and coil inductance $L_{\mathrm{coil}}$ and coil resistance $R_{\mathrm{coil}}$ related to a spiral coil within the forming die. Similarly, in Figure 2, an electrically conductive workpiece in the forming die is expressed as an electric circuit composed of workpiece inductance $L_{w}$ and workpiece resistance $R_{w}$. In Figure 2, *U* is charging voltage, I(t) is an input current discharged by *U*, and $I_{w}\left(t\right)$ is a current flowing in a workpiece induced by a magnetic field resulting from I(t).

The EMF apparatus in Figure 2 can be characterized as a low-inductance, large-capacitance electric circuit with a high current switch that triggers a high-frequency current pulse to the spiral coil in the forming die. The flick of the high current switch releases an extremely high current in a very short time period, which causes the coil to induce an intense magnetic field around a workpiece in the forming die by Faraday’s law of induction. As a result, the resultant electric and magnetic force together lead to the Lorentz force, which pushes the conductive workpiece away from the forming die to instantaneously deform it [20]. To generate an input current I(t) for EMF, we set an initial charge voltage $U\left(0\right)=U_{0}$ to be 9 kV in the control system, and capacitance *C* and equivalent inductance *L* measured 333 μF and 5.426 μH, respectively. With the help of the Rogowski coil and the oscilloscope in Figure 3, we measured and monitored the input current I(t) as plotted in Figure 4; the maximum I(t) of 58 kA occurred at 48.4 μs, and the current frequency was 4.54 kHz, i.e., $2.2\times 10^{-4}$ s in terms of the current pulse. The energy capacity of the EMF apparatus at the charging voltage of 9 kV was found as 13.4 kJ by $E=\frac{1}{2}CU_{0}^{2}$.

#### 3.1.2. Experimental Apparatus

Figure 5b delineates a closer look of the forming die; Figure 1 and Figure 5e show the spiral coil in the forming die along with the insulation block. As an illustration, Figure 5a depicts the explanatory diagram of the forming die in Figure 5b. In Figure 5e, the spiral coil is embedded in the insulation block made of epoxy, and the block is protected from expansion and fracture by the aluminum bandage. The forming die in Figure 5b has four holes for inserting bolts to combine the spiral coil with the upper die, and the coil is connected to the EMF apparatus via two leads shown in Figure 5a. For the clamping force, 71,875 N was applied by adding 64.68 N of the coil’s self-weight, 440 N of the upper mold’s self-weight, and 71,370 N of the proof load of the bolt. This clamping force functioned as the holding force in the experiment and was used as a blank holder force in the simulation.

### 3.2. Electromagnetic Forming Simulation

#### 3.2.1. Coupled Finite/Boundary Element Model

For the numerical analysis of the EMF process, we developed an electromagnetic-structural coupling simulation using LS-DYNA with the EM module. Normally, the EMF simulation is a complex, time-consuming process because the electromagnetic analysis of surrounding air/insulators with a finite element method (FEM) necessitates a considerably large mesh compared to the nonlinear structural analysis of a solid conductor. To address this computational issue, LS-DYNA with the EM module utilizes a boundary element method (BEM) instead of an FEM to avoid meshing the entire domain of surrounding air/insulators. For coupled electromagnetic and structural analysis, LS-DYNA with the EM module first applies a BEM to the Maxwell equations with eddy-current approximation to analyze the electromagnetic field of surrounding air/insulators. Next, LS-DYNA evaluates the Lorentz force at the boundaries of a solid conductor and employs an FEM to predict the nonlinear structural deformation of a solid conductor [21,22].

To simulate the free bulge test with LS-DYNA, we constructed the computational models of a spiral coil, an insulator, a sheet, and a die as depicted in Figure 6. The spiral coil part in Figure 6 is delineated from the top view in Figure 7 for the illustration of the overall shape. The spiral coil was made of copper and sized by a maximum diameter of 110 mm. We fabricated the coil with a 5 mm by 10 mm rectangular strip after turning it six times. The workpiece is a flat square sheet made of aluminum 1050-H14 with each side and the thickness of the sheet measuring 160 mm and 1 mm, respectively.

For finite element (FE) analysis, we modeled the spiral coil with 5051 solid elements, the insulator with 2482 solid elements, the sheet with 8508 solid elements (three elements through the thickness to capture the bending effect), and the die with 3707 shell elements. Note that the die, the coil, and the insulator were treated as rigid bodies without plastic deformation. In total, we discretized the assembly of the four parts into 19,748 elements and 24,128 nodes for FE analysis. We found that the chosen element and node sizes resulted in accurate EMF simulation from the previous study [23]. To initiate EMF simulation, we used the measured current depicted in Figure 4 for the input current and simulate for 700 μs. To take account of the friction between the sheet and the mold, we set the friction coefficient to 0.15. The electrical and structural material properties adopted for the coupled structural-electromagnetic simulation are listed in Table 1. Note that electrical resistivity is an important factor because it affects the electrical conductivity of materials. To couple structural deformation analysis with electromagnetic analysis by a BEM, we evaluated the Lorentz force *F* as the cross-product of a current density *J* and a magnetic flux density *B* such that F=J×B.

#### 3.2.2. Constitutive Model

In general, static flow stress can be expressed as: (25)$$\bar{\sigma }=K\epsilon^{n}$$
where σ is true stress, ϵ is true strain, and both *K* and *n* are material dependent parameters determined by a tensile test. Since the static constitutive model in Equation (25) cannot allow for strain rates, we are required to draw on constitutive models suitable for high-speed forming, such as Johnson–Cook [24], Zerilli–Armstrong [25], and Steinberg [26] models. For the choice of a constitutive model, we employed the modified Johnson–Cook material model [21] defined by: (26)$$\sigma_{y}=\left[A+B{\left(\epsilon_{\mathrm{eff}}\right)}^{n}\right]{\left[1+C\left(\ln\frac{\dot{\epsilon }}{\epsilon_{0}}\right)\right]}^{p}$$
where *A*, *B*, and *n* are parameters acquired with quasi-static tensile test data, $\epsilon_{\mathrm{eff}}$ is equivalent plastic strain, $\dot{\epsilon }$ is a strain rate at a dynamic state, and $\epsilon_{0}$ is a reference strain rate obtained from a quasi-static tensile test. In the case of aluminum 1050-H14, we found that *A* is 111 MPa, *B* is 38.2 MPa, *n* is 0.2548 with an MTS Landmark test machine, and $\epsilon_{0}$ is obtained at 0.00667 /s. In Equation (26), both *C* and *p* are two unknown constitutive parameters that account for the effect of high strain rates, and will be determined with the developed LS-DYNA simulation model via inverse identification. To adopt the constitutive model in Equation (26) for the LS-DYNA simulation, we used the piecewise linear plasticity material model, MAT-24, which interpolates a set of flow stress data at prearranged strain rates to predict flow stress data at an arbitrary strain rate [22]. To this end, we set the range of strain rates from 0.1 /s to 6000 /s and evaluated flow stress data at a total of eight strain rates: 0.1 /s, 1 /s, 10 /s, 100 /s, 1000 /s, 2000 /s, 4000 /s, and 6000 /s.

## 4. Electromagnetic Forming Simulation Reduction

In this section, we deal with constructing and verifying a reduced EMF simulation for rapid, approximate evaluation to expedite the inverse identification process. For the MOR of EMF simulation, we utilized PCA to extract an empirical orthonormal basis and Kriging to predict changes of basis coefficients.

### 4.1. Reduced Model Construction

To facilitate inverse constitutive modeling with a rapid forming simulation, we applied MOR consisting of PCA and Kriging to the EMF simulation of a free bulge test, implemented by LS-DYNA with the EM module. Here, we briefly delineate the overall process of MOR utilized to accelerate the EMF simulation for the sake of completeness. For more information or details regarding other model approximation techniques explored for the EMF simulation, please refer to Refs. [23,27].

To construct and verify a reduced EMF simulation, we populated training and testing data sets with Latin hypercube and uniform random designs, respectively, as shown in Figure 8. Each of the training and testing data sets comprises 20 samples of the two constitutive parameters of the modified Johnson–Cook model, *C* and *p*. For the sample data generation, we set the parameter domain of interest as $D_{\theta }=\left[0.001,1.0\right]\times \left[0.012,2.2\right]\subset R^{2}$, which is conjectured to encompass the yet-to-be-found constitutive parameters based on the authors’ intuition and experience. Note that a caveat should be associated with this parameter-bound selection because otherwise we may end up carrying out computationally expensive forming simulations many more times than needed after repeatedly adjusting the bounds. Once we collected the EMF simulation output at the training samples, we invoked PCA upon the compiled EMF simulation output to empirically extract an orthonormal basis.

As an illustration, Figure 9 exhibits the first two dominant basis vectors of the coordinates of the deformed configurations, characterizing the EMF simulation output obtained with the training samples. The basis vectors in Figure 9 are not perfectly symmetric with respect to the middle of the coordinates because of slight asymmetricity in the spiral coil and human fallibility in discretizing and tabulating the EMF simulation output. As shown in Figure 9, the first basis vectors delineate macroscopic bulge deformation in the central region, whereas the second basis vectors depict microscopic fluctuating deformation near the flanges. In Figure 9, the magnitudes of the first and second basis vectors are alike because each basis vector is normalized to be a unit vector. However, the influences of the first and second basis vectors on deformation are unlike because the coefficient of the first basis vector is an order larger than that of the second basis vector (as conveyed in Figure 10 in Section 4.1). As a result, the first basis vector is related to major deformation in the middle, and the second basis vector is associated with minor deformation around the edge. Note that the first basis vectors in Figure 9 are drawn upside down portraying the bulge deformation in the opposite direction, which is plausible due to the sign ambiguity of eigenvectors.

Following the basis evaluation, we examined eigenvalues corresponding to eigenvectors, i.e., basis vectors, to decide the proper size of a reduced EMF simulation. According to the PCA formulation in Section 2.2.2, Equations (17) and (19) convey that an eigenvalue denotes variation explained by the associated eigenvector. Thus, we evaluated the cumulative sum of eigenvalues normalized by the total eigenvalue sum to quantify the significance of each basis vector. From the eigenvalue analysis, it turned out that we could capture 99.84% variation observed in the EMF simulation output with the first basis vector alone, and together with the second basis vector, we can account for variation in the EMF simulation output up to 99.99%. Therefore, we chose the first two leading basis vectors to build a reduced EMF simulation.

Once we settled on the basis size for MOR, we evaluated basis coefficients multiplied by the two basis vectors with the compiled EMF simulation output to relate each basis coefficient to the two constitutive parameters. For the Kriging model construction, we utilized ooDACE [28], which provides various Kriging implementations in MATLAB. In using ooDACE, we located the optimal hyperparameters with the genetic algorithm (GA), which comes with the global optimization toolbox in MATLAB. For the GA optimization process, we set both the lower and upper search bounds of hyperparameters to $10^{-3}$ and $10^{3}$, respectively, and set the population and generation sizes to 50 and 100, respectively.

After the Kriging model construction, we found the optimal parameter estimates of each Kriging model as listed in Table 2. Since a larger hyperparameter implies a more influential input, Table 2 shows that *C* is more dominant than *p* in estimating the first basis coefficient $a_{1}$. Similarly, *p* affects the second basis coefficient $a_{2}$ marginally more than *C*. Based on the parameter estimates in Table 2, Figure 10 illustrates the contours of the two basis coefficients generated by the two Kriging models. Note that the magnitude of $a_{1}$ is greater than that of $a_{2}$ in Figure 10 because the first basis vector weighted by $a_{1}$ should dictate most of thee bulge deformation observed in the EMF simulation output according to the eigenvalue analysis.

### 4.2. Reduced Model Verification

To verify the constructed reduced EMF simulation, we first investigated the prediction accuracy of the two Kriging models and then that of the reduced EMF model. To begin with the Kriging models, we inspected prediction accuracy only with the testing data because Kriging interpolation is precise at the known training data by formulation.

First, Table 3 shows the numerical verification results of the two Kriging models, measured in terms of a coefficient of determination ($R^{2}$) and a mean absolute relative error (MARE); the closer the $R^{2}$ and MARE values are to one and zero, respectively, the closer the predicted data are to the actual data. Next, Figure 11 shows the graphical verification results of the two Kriging models, comparing basis coefficients estimated by the Kriging models to their exact counterparts evaluated by the inner product of the EMF simulation output and the basis vectors; the more scattered data coincide with the diagonal line, the closer the estimated data are to the exact data. As substantiated by Table 3 and Figure 11, the two Kriging models were able to accurately estimate basis coefficients. Between the first and second basis coefficients, the estimates of the first basis coefficient are found to be more accurate compared to those of the second basis coefficient. However, this inferior accuracy of the second basis coefficient does not concern the prediction capability of the reduced EMF simulation because the contribution of the second basis coefficient is inconsequential, about 0.15%, as revealed by the eigenvalue analysis.

After examining the two Kriging models, we repeated the same numerical and graphical verification processes with the reduced EMF simulation using both the training and testing data sets. First, Table 4 shows the numerical verification results of the reduced EMF simulation, summarizing $R^{2}$ and MARE values obtained with each data set. In Table 4, the averages of *R*^2^s are nearly one regardless of the data sets, and those of MAREs are about 10% and 20% in the case of the training and testing data sets, respectively.

Next, Figure 12 presents the graphical verification results of the reduced EMF simulation, depicting the output predicted by the reduced EMF simulation with respect to that generated by its original EMF simulation in the case of the testing data. Figure 12 clearly shows that the predicted output of the reduced EMF simulation almost accords with the actual output of the original EMF simulation. Overall, Table 4 and Figure 12 substantiate that the prediction capability of the reduced EMF simulation is quite acceptable and reliable.

Although the numerical verification results in Table 4 assure the prediction capability of the reduced EMF simulation, the maximums of the MAREs are relatively large. To further explore sources of errors causing large MAREs in Table 4, we evaluated *R*^2^s with respect to each coordinate of the deformed shape. As shown in Figure 13, most discrepancies arise near the flanges rather than the central region. Since the changes of the two constitutive parameters mostly generate differences in the central region, we surmise that the prediction errors occurring near the edges would not jeopardize inverse parameter estimation.

## 5. Inverse Identification Results

In this section, we embark on indirectly identifying the two constitutive parameters of the modified Johnson–Cook model in relation to the EMF simulation of a free bulge test with the help of MOR. Given the measurements of an actual workpiece, we estimated the constitutive parameters by means of regularized nonlinear least squares in which the optimal regularization parameter was determined by the L-curve method. After inverse parameter estimation, we verified the reduced EMF simulation and then validated the original EMF simulation at the identified parameters.

### 5.1. Experimental Measurements

For the experiment with the EMF apparatus in Figure 5, we used a specimen made of aluminum 1050-H14; it was shaped as a square sheet whose sides and thickness measured 160 mm and 1 mm, respectively. We placed the specimen in-between the forming coil and the lower die in Figure 5c. Because of the weight of the upper part of the die and torque applied to the pillars of the die, the specimen was subject to 0.11 MPa of uniform force. After setting up capacitor charging and die clamping, we let an input current flow through the spiral coil, which resulted in the deformed workpiece as shown in Figure 14. For measurements, we repeated the free bulge test three times to obtain a deformed workpiece as shown in Figure 14. We used a 3-D scanning apparatus to gauge the cross-section of the deformed configurations across the (X, Z) plane as depicted in Figure 15, where the maximum deflection is 26.61 mm. Note that the inverted bell-shaped measurements in Figure 15 are slightly non axisymmetric with respect to the vertical center line because of non-uniform magnetic force resulting from the asymmetric spiral coil.

### 5.2. Parameter Estimation

To overcome the ill-posed nature of inverse parameter estimation, we leveraged a regularization parameter that makes the misfit function smoother during parameter search. We addressed the regularized nonlinear least squares problem posed in Equation (7) with constrained nonlinear optimization implemented by fmincon [29] in MATLAB. For the optimization, we chose sequential quadratic programming (SQP) [30] as an optimizer and set both function and step tolerances to 1.0 $\times \;10^{-9}$. In addition, we doubled the maximum number of function calls to 400 to ensure convergence to the local minimum. Each optimization process is initiated with a random initial point generated by a uniform distribution whose range was set by the constitutive parameters bound $D_{\theta }$ in Section 4.1 for the reduced EMF simulation being valid throughout optimization.

To determine the optimal regularization parameter with the *L*-curve method, we swept through 1000 samples of the regularization parameter evenly spaced between $10^{-2}$ and $10^{2}$ in a log scale. To wit, we repeatedly solved the regularized nonlinear least squares problem in Equation (7) for 1000 times at each different regularization parameter. After 1000 runs of the optimization, we confirmed that each optimization run had reached the local minimum. Note that this series of successive optimization runs inevitable for the *L*-curve method aggrandizes the benefit of rapid simulation via MOR. For instance, the regularization parameter sweep required 111.2 and 111,198 function calls on average and in total, respectively. Considering an EMF simulation by LS-DYNA took approximately 4 h on a workstation, (The EMF simulation implemented by LS-DYNA operated on Windows 7 Pro equipped with two Xeon 3.10 Ghz CPUs and 2 GB RAM.) it would require about 50.7753 years if we would have addressed the same problem without MOR. Instead, we spent about 6.6667 days for the 40 simulation runs with the training and testing data sets in total, as the reduced EMF simulation ran instantaneously.

As a result of nonlinear least square with regularization parameter sweep, we obtained the *L*-curve as shown in Figure 16, which delineates a series of residual and solution norms at various regularization parameters. In Figure 16, the optimal regularization parameter is marked at the kink of the *L*-curve with the red circle, and the two constitutive parameter estimates corresponding to the optimal regularization parameter are listed in Table 5. To further explore the results of inverse parameter estimation with the *L*-curve, we draw the contours of residuals between predictions and the measurements in Figure 17, where the trails of black dots are constitutive parameter estimates corresponding to the regularization parameters in Figure 16. Note that Figure 17 shows a flat, diagonal valley that represents a myriad of possible parameter estimates, which reveals uncertainty in parameter estimates that is common to an inverse problem; without regularization, parameter estimates may get trapped anywhere within the valley.

### 5.3. Simulation Model Verification and Validation

After inverse parameter estimation, we are required to validate the original EMF simulation at the identified constitutive parameters to see if they actually cause the original EMF simulation to predict a deformed configuration that matches the given experimental measurements. Before validation, we first verified the reduced EMF simulation against the original one at the identified parameter estimates because, in general, we cannot assure the accuracy of the reduced EMF simulation at parameter values other than those in the training and testing data sets.

For the verification of the reduced EMF simulation, Figure 18 compares the output of the reduced EMF simulation to those of the original one evaluated at the identified constitutive parameters. As shown in Figure 18, the outputs of the reduced EMF simulation, depicted by blue dashed lines with squares, closely align with those of the original EMF simulation, delineated by red solid lines with dots. In addition, $R^{2}$ and MARE values evaluated with the results in Figure 18 were found to be 9.999524 $\times \;10^{-1}$ and 1.467169 $\times \;10^{-1}$, respectively, which are acceptable in comparison to the numerical verification results in Table 4. Therefore, we confirmed that the reduced EMF simulation was able to yield the same output as the original counterpart at the identified parameter estimates.

Followed by the verification of the reduced EMF simulation, we validated the output of the original EMF simulation with respect to the actual measurements obtained from the workpiece manufactured by a free bulge test with EMF. As with Figure 18, the numerical predictions of the original EMF simulation are depicted by red solid lines with dots, and the three sets of measurements are drawn with black circles in Figure 19. Unlike the verification results in Figure 18, the validation results in Figure 19 exhibit slight deviations of the numerical predictions from the experimental measurements. In particular, the discrepancies are mostly noticeable near the peak and the right hillside of the bulge deformation. Numerical examination found the averaged $R^{2}$ and MARE values as 9.945031 $\times \;10^{-1}$ and 2.009067 $\times \;10^{-1}$, respectively, which are slightly worse than the previous verification results.

The marginally underestimated peaks in Figure 19 have been observed in other EMF simulations of free bulge tests performed by the authors and require further effort from a validation side. Despite minor discrepancies, Figure 19 shows that the numerical simulation at the identified constitutive parameters may yield the deformed shape, which agrees considerably well with that of the actual workpiece.

## 6. Validation with a Dynamic Material Test

In this section, we validate the previously identified constitutive model with a dynamic material test using an SHPB to examine the utility and accuracy of the inverse identification strategy.

### 6.1. Experimental Apparatus

To measure dynamic stress-strain data of aluminum 1050-H14, we utilized the in-house built SHPB apparatus shown in Figure 20. Schematically, the apparatus in Figure 20 can be illustrated as an assembly of strike, incident, and transmitted bars along with a data acquisition (DAQ) system as depicted in Figure 21. For a dynamic material test, a specimen is placed in-between an incident bar and a transmitted bar. When the strike bar impacts the end of the incident bar at a very high speed, a stress wave called the incident wave is emitted in the incident bar and transmits through the bar towards the specimen. Immediately upon the incident wave arriving at the specimen, it splits into two stress waves: the transmitted wave and the reflected wave. The former propagates through the specimen due to impedance difference between the incident bar and the specimen. As a result, plastic deformation in the specimen occurs until the transmitted wave contacts the transmitted bar. The latter reflects away from the specimen and goes back to the incident bar. The incident and transmitted waves are picked up by two strain gauges, each of which adheres to the incident and the transmitted bars, and relayed to a DAQ system by two separate channels.

For experimental data processing, we utilized the in-house built DAQ system composed of two strain gauges, a signal conditioning amplifier, an oscilloscope, and a personal computer (PC). During the experiment, plastic deformation causes changes of electrical resistance in the strain gauges, which transform to voltage with the help of a Wheatstone bridge. Afterward, the conditioning amplifier enhances voltage signals from the strain gauges, and then the oscilloscope records voltage fluctuation with respect to time. Last, the PC converts the voltage data into stress-strain data. The DAQ system used for this research has a high frequency response enough to transcribe signals from strain gauges, as a dynamic material test finishes in less than a millisecond. Likewise, the signal conditioning amplifier employed for this research can deal with frequency bandwidths up to 125 kHz.

To calculate the impact speed of a strike bar, we utilized a photogate device typically used for precise measurements of high-speed events by means of optical sensing. The photogate used for this research has two gates 100 mm apart as shown in Figure 22, and the measurement mechanism is illustrated in Figure 23. As depicted in Figure 23, a photogate senses an event time when an object blocks light beamed in the gates. In our case, the photogate in Figure 22 measured two event times when the strike bar passed through the first and second gates successively. Since the gate distance as well as the time difference are available, we can easily compute the average speed of the strike bar. In our case, the impact speed measured 47 m/s, and Figure 24 shows the specimen before and after the SHPB test on the left and right sides, respectively.

### 6.2. Validation Results

With the two strain gauges on the pressure bars, we obtained stress waves in the form of voltage as plotted in Figure 25, out of which incident strain, reflective strain $\epsilon_{R}$, and transmitted strain $\epsilon_{T}$ were evaluated. From these strain data, we computed engineering stress $\sigma_{s}$, engineering strain $\epsilon_{s}$, and the strain rate $\dot{\epsilon_{s}}$ based on Equations (27)–(29), respectively, as follows: (27)$$\sigma_{s}=E\left(\frac{A_{\mathrm{bar}}}{A_{s}}\right)\epsilon_{T}$$
(28)$$\epsilon_{s}=\frac{-2C_{0}}{L_{s}}\int_{0}^{t}\epsilon_{R}\left(t\right)dt$$
(29)$$\dot{\epsilon_{s}}=\frac{-2C_{0}}{L_{s}}\epsilon_{R}\left(t\right)dt$$
where *E* is the elastic modulus of a specimen, $A_{\mathrm{bar}}$ and $A_{s}$ are the sectional areas of an incident bar and a specimen, respectively, $L_{s}$ is the length of a specimen, and $C_{0}$ is the transmission speed of elastic wave in the strike bar. For illustration, the measured strain rate is depicted in Figure 26, where the maximum strain rate is about 4500 /s at a time period of 0.00002 s to 0.00007 s. Note that the measured strain rates are negative in Figure 26 because of compression.

As a result of the SHPB test, we attained a stress-strain curve of aluminum 1050 H-14 at a strain rate of 4500 /s as shown in Figure 27 with the SHPB test. For a comparative study, we also presented flow stress data at the same strain rate numerically obtained by the inverse identification along with those at a strain rate of 0.00667/s achieved by a quasi-static tensile test. In Figure 27, the black, red, and blue lines indicate the stress-strain data of the SHPB test, the inverse identification, and the quasi-static tensile test, respectively. In Figure 27, the yield point of flow stress data with the SHPB test is about twice as high as that with the quasi-static tensile test because of work hardening. As the stress levels of flow stress data with the SHPB test are a little larger than those with the inverse identification, work hardening seems a little underestimated in the case of the inverse identification compared to the SHPB test. Overall, flow stress data numerically identified are quite comparable to those experimentally achieved. Since the experimental approach is more resource-intensive than the numerical approach, we believe that the numerical approach is more accessible and affordable.

## 7. Conclusions and Future Work

We formulated constitutive parameter estimation in the context of high-speed forming simulation in the form of nonlinear least squares with regularization. Since the regularization parameter is crucial, we employed the *L*-curve scheme to properly select the regularization parameter, which optimally balances the residual and regularization terms. As a consequence of the *L*-curve, a stark increase in computational cost is unavoidable due to regularization parameter exploration. To cope with this computational burden, we leveraged MOR to convert an accurate, time-consuming forming simulation into an approximate, rapid substitute.

As proof of concept, we utilized the proposed approach to determine the two constitutive parameters of the modified Johnson–Cook model for the EMF simulation of a free bulge test with the workpiece made of aluminum 1050 H-14. The EMF simulation was implemented by LS-DYNA and then approximated by MOR composed of PCA and Kriging techniques. Thanks to the reduced EMF simulation, inverse parameter estimation was done swiftly, and tremendous computational saving was achieved roughly from 50.7753 years to 6.6667 days by a factor of approximately 2780. For accuracy investigation, we carried out the verification of the reduced EMF simulation and the validation of the original EMF simulation at the identified constitutive parameters, and the verification and validation results confirmed reasonable consistency between numerical simulation models and the manufactured workpiece. Furthermore, the numerically identified constitutive model was found to be accurate compared to flow stress measurements from the dynamic material test.

In conclusion, we substantiated the utility of the regularized nonlinear least squares formulation benefited from MOR with a demonstration of the Johnson–Cook parameter estimation for an EMF simulation. To summarize, we accomplished the following contributions through this research: (i) the generalized formulation of inverse constitutive parameter estimation, suitable for any high-speed forming simulation; (ii) the use of regularized nonlinear least squares with the *L*-curve method for effective inverse parameter estimation; (iii) the adoption of a rapid approximation of a forming simulation for computational feasibility and viability; and (iv) the demonstration of the proposed strategy for constitutive parameter identification with a free bulge test using EMF.

For future work, we would like to see more applications of the suggested approach for inverse parameter identification with various constitutive models in connection with other high-speed forming techniques, such as EHF. We would also like to tackle the same problem of inverse parameter estimation for high-speed forming simulation from a probabilistic perspective. This non-deterministic perspective may allow us to systematically deal with uncertainties associated with not only an approximate simulation model but also experimental measurements. Moreover, this perspective change would enable us to quantify uncertainties in constitutive parameter estimates, which would eventually affect uncertainties in flow stress predictions.

## Figures and Tables

**Figure 1 materials-15-07179-f001:**
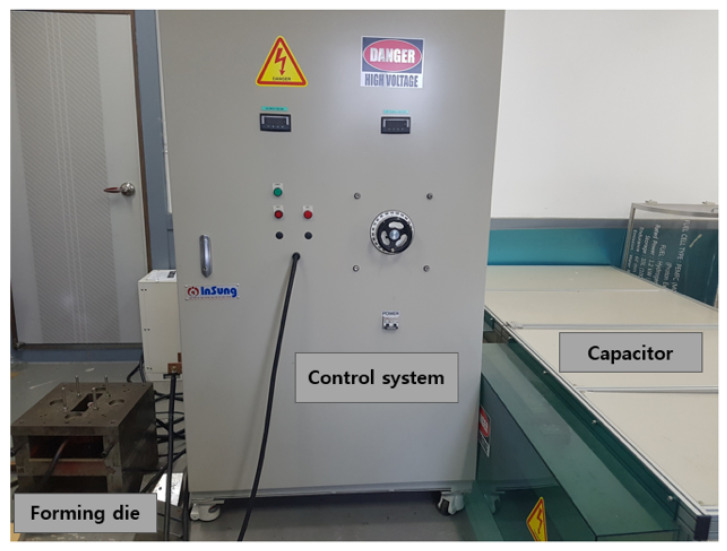
EMF equipment (PNU-32).

**Figure 2 materials-15-07179-f002:**
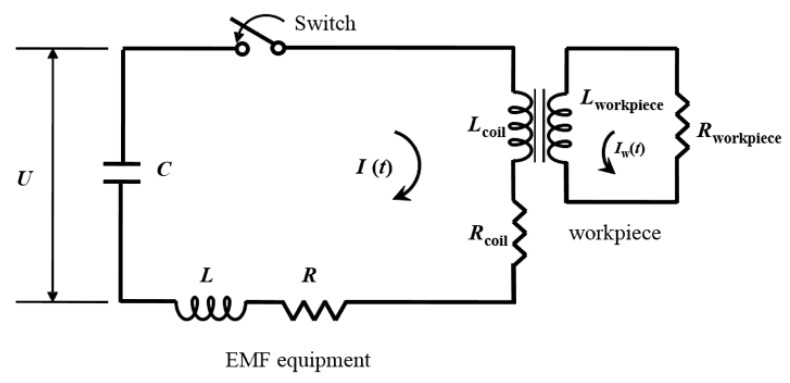
RLC equivalent circuit diagram of the EMF equipment and a workpiece.

**Figure 3 materials-15-07179-f003:**
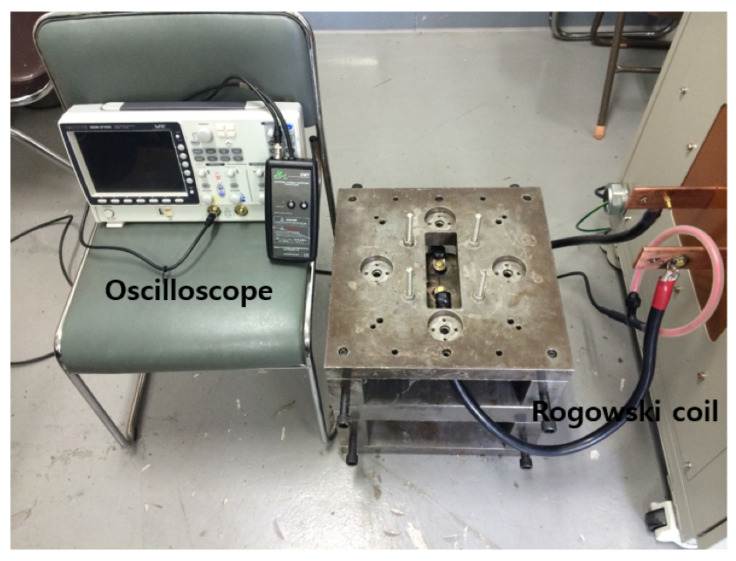
Input current measurement with a Rogowski coil and monitoring with an oscilloscope.

**Figure 4 materials-15-07179-f004:**
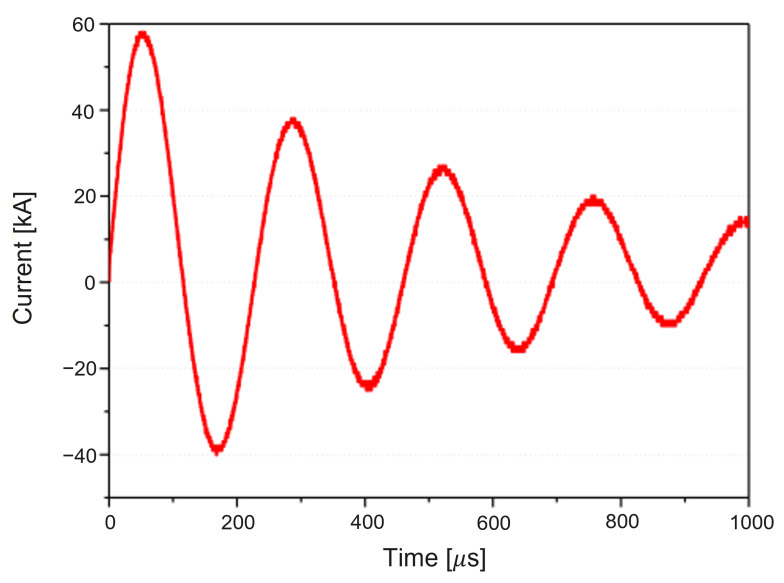
Measured input current at a charging voltage of 9 kV.

**Figure 5 materials-15-07179-f005:**
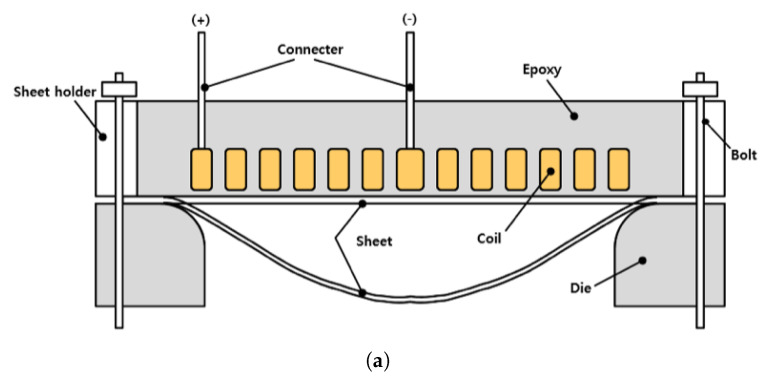
Experimental set up for EMF free bulge test. (**a**) Forming coil with a test bed. (**b**) Forming die. (**c**) Cross-section of a forming die. (**d**) Forming coil. (**e**) Insulation block.

**Figure 6 materials-15-07179-f006:**
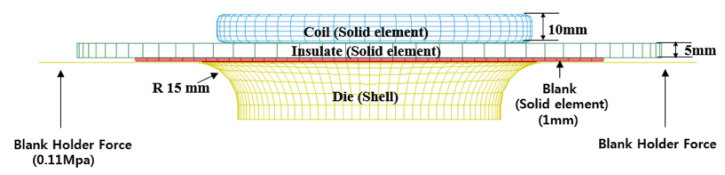
Computational models for a free bulge test with EMF.

**Figure 7 materials-15-07179-f007:**
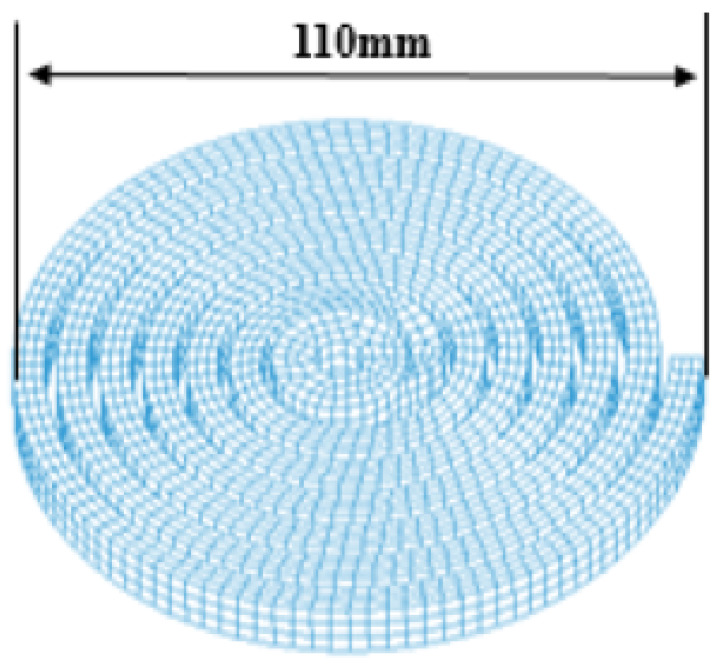
Forming coil model.

**Figure 8 materials-15-07179-f008:**
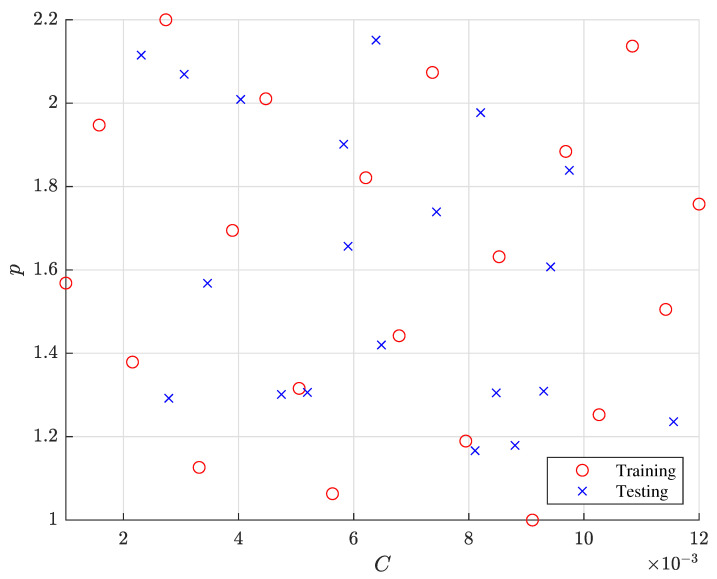
Sampled constitutive parameters in the training and testing data sets.

**Figure 9 materials-15-07179-f009:**
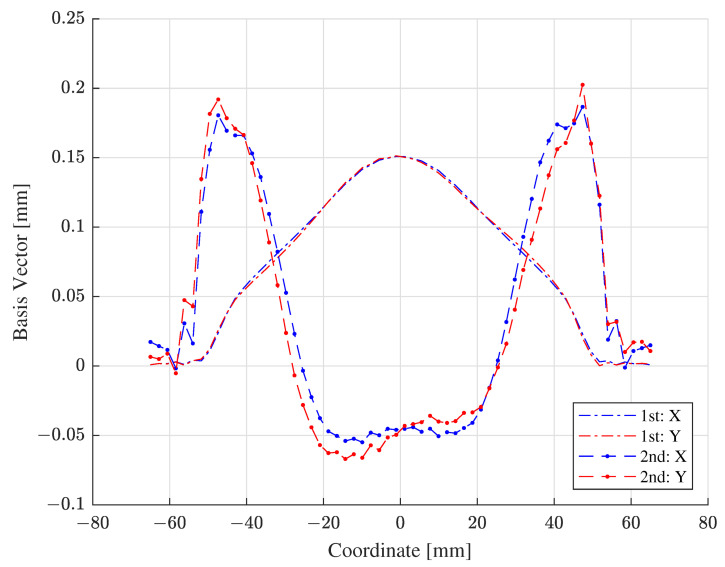
The 1st and 2nd basis vectors of the EMF simulation output.

**Figure 10 materials-15-07179-f010:**
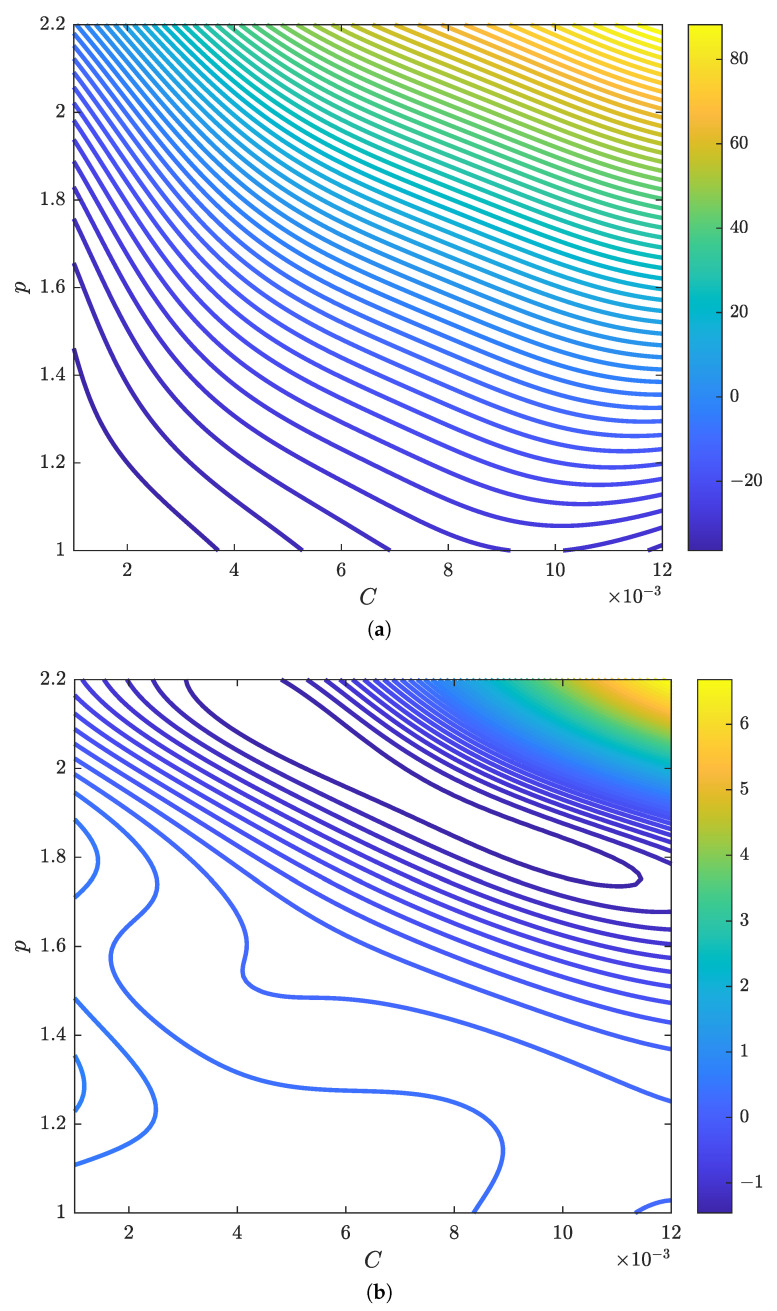
Contours of the first two basis coefficients estimated by the constructed Kriging models. (**a**) 1st coefficient $a_{1}$. (**b**) 2nd coefficient $a_{2}$.

**Figure 11 materials-15-07179-f011:**
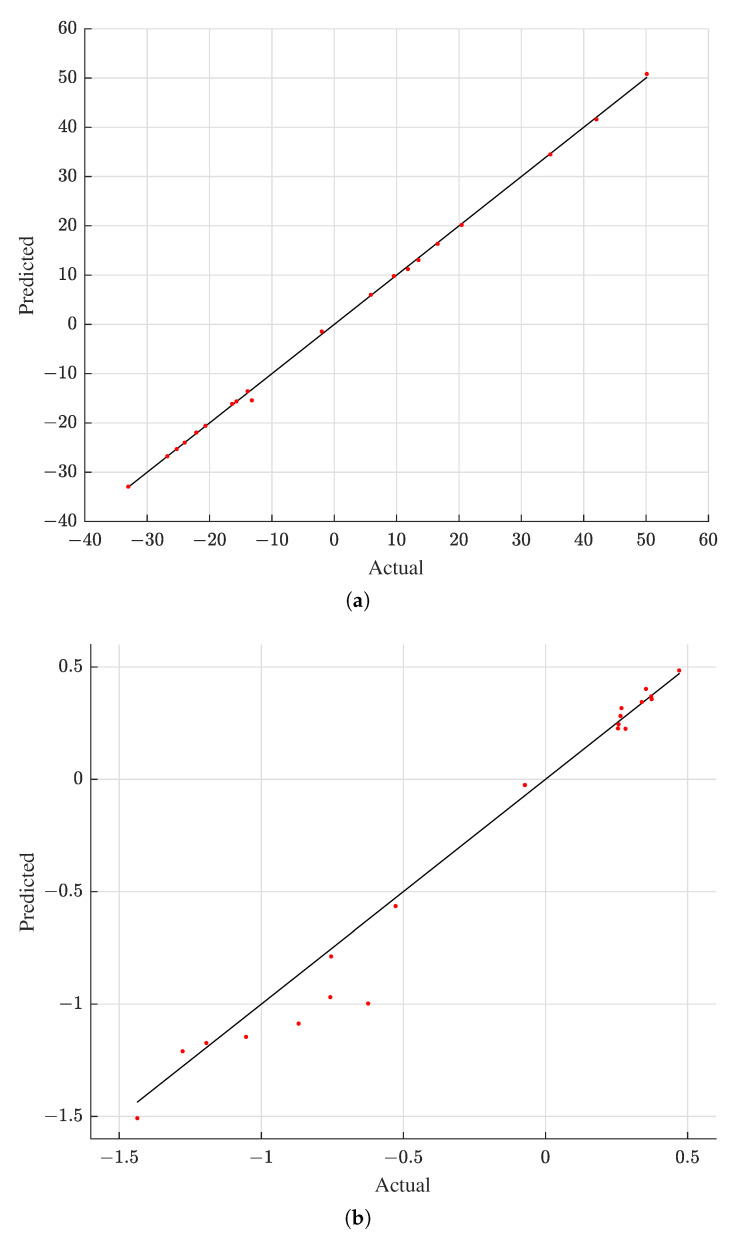
Actual versus predicted basis coefficients with the testing data. (**a**) 1st coefficient $a_{1}$. (**b**) 2nd coefficient $a_{2}$.

**Figure 12 materials-15-07179-f012:**
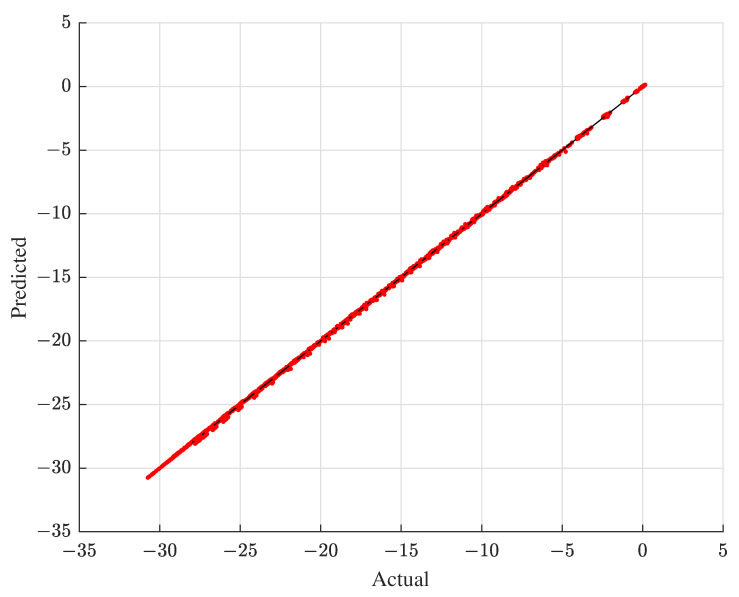
Actual versus predicted EMF simulation output with the testing data.

**Figure 13 materials-15-07179-f013:**
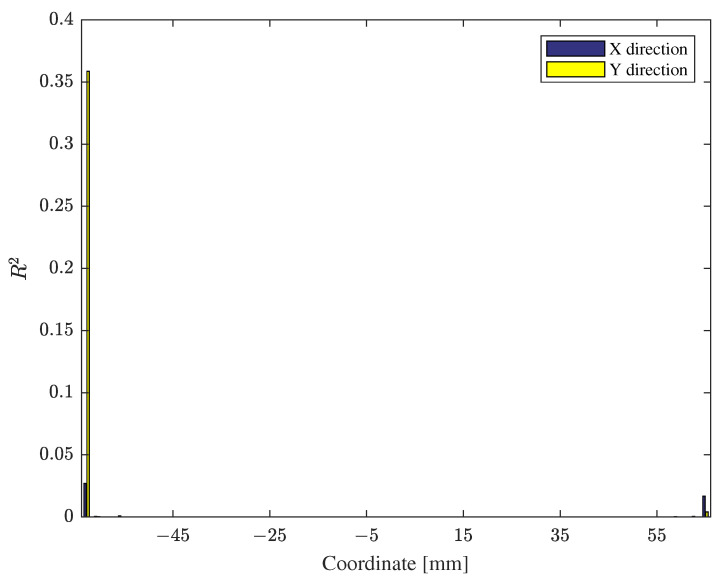
$R^{2}$ values of the predicted EMF simulation output measured across coordinates in X and Y directions with the testing data.

**Figure 14 materials-15-07179-f014:**
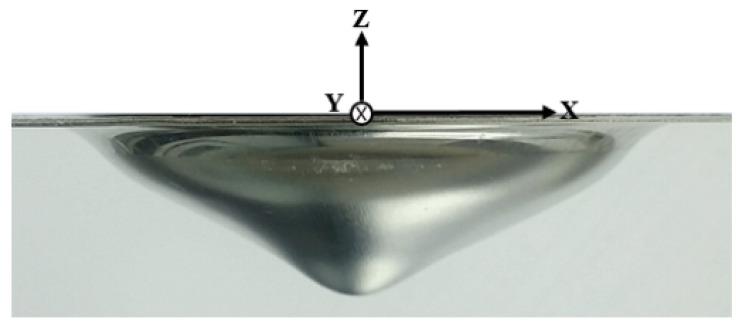
Deformed sheet after the free bulge test.

**Figure 15 materials-15-07179-f015:**
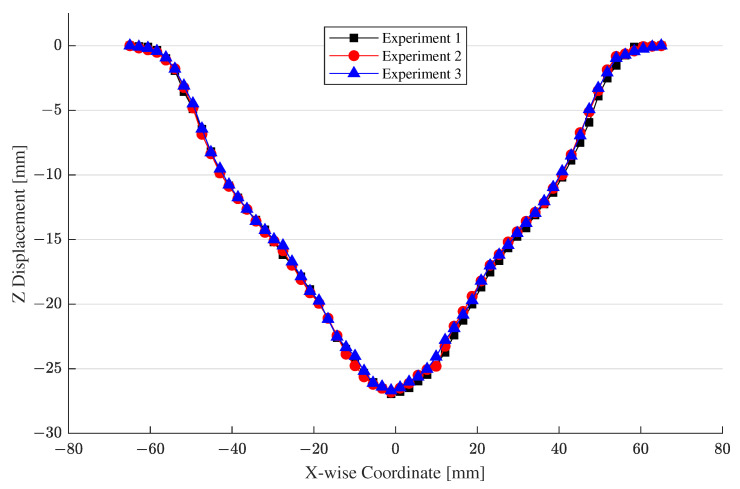
Measurements of the deformed sheet.

**Figure 16 materials-15-07179-f016:**
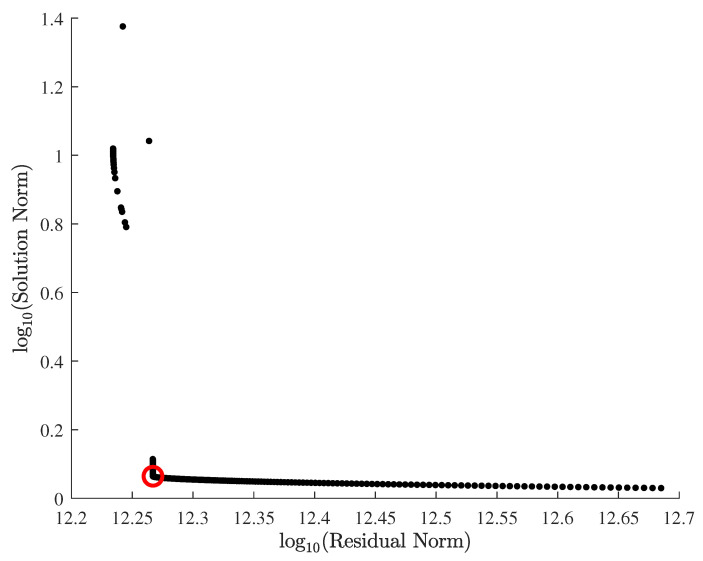
*L*-curve for the inverse estimation of the modified Johnson–Cook constitutive parameters.

**Figure 17 materials-15-07179-f017:**
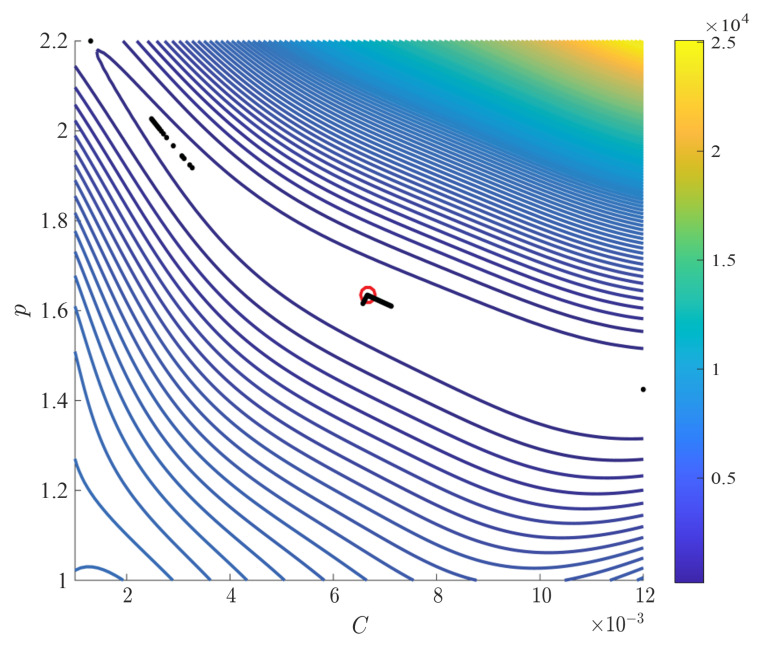
Contours of residuals between predictions and the measurements with constitutive parameter estimates found by the *L*-curve method.

**Figure 18 materials-15-07179-f018:**
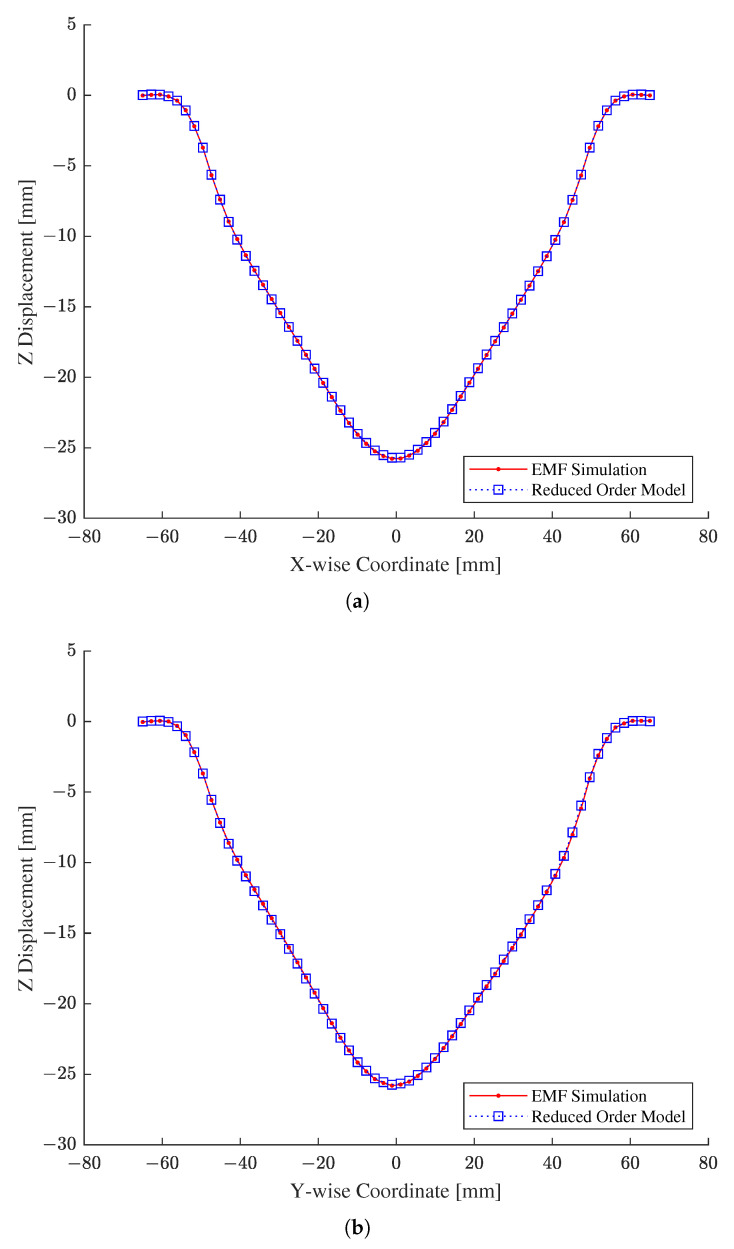
Verification of the constructed reduced order model with respect to the original EMF simulation. (**a**) X-wise coordinates. (**b**) Y-wise coordinates.

**Figure 19 materials-15-07179-f019:**
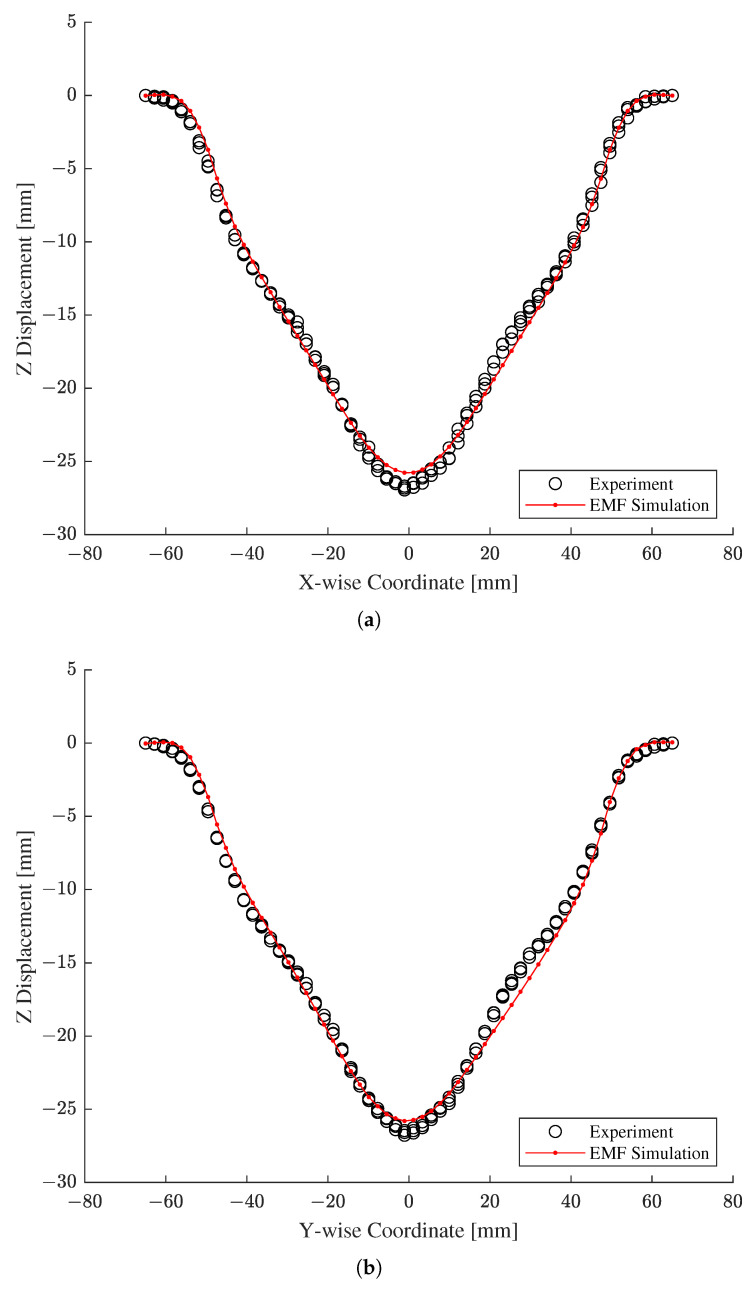
Validation of the predictions of the original EMF simulation with respect to the actual measurements. (**a**) X-wise coordinates. (**b**) Y-wise coordinates.

**Figure 20 materials-15-07179-f020:**
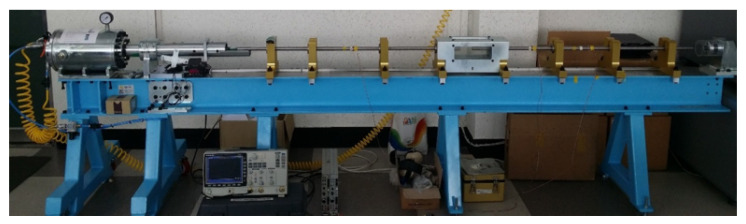
SHPB equipment.

**Figure 21 materials-15-07179-f021:**
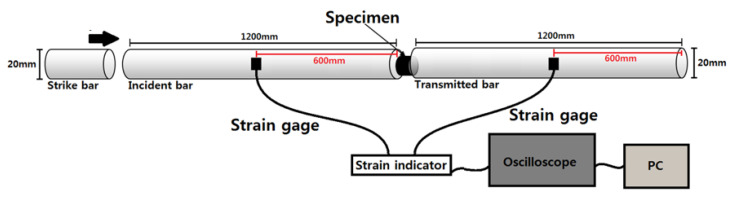
Schematic of the SHPB equipment with a DAQ system.

**Figure 22 materials-15-07179-f022:**
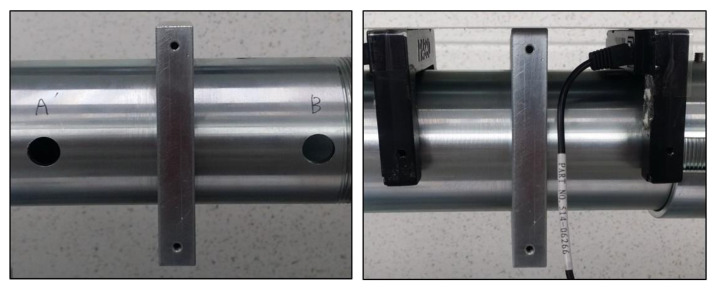
Photogate installed on the SHPB apparatus.

**Figure 23 materials-15-07179-f023:**
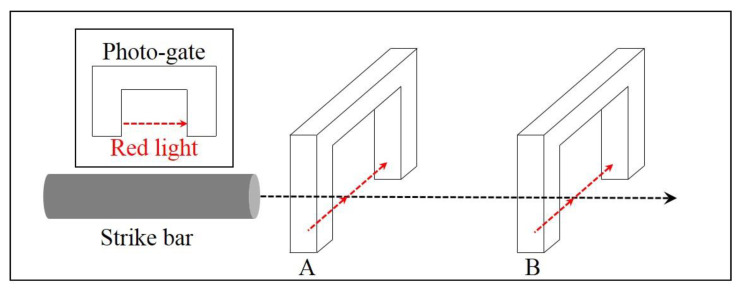
Schematic illustration of photogate mechanism.

**Figure 24 materials-15-07179-f024:**
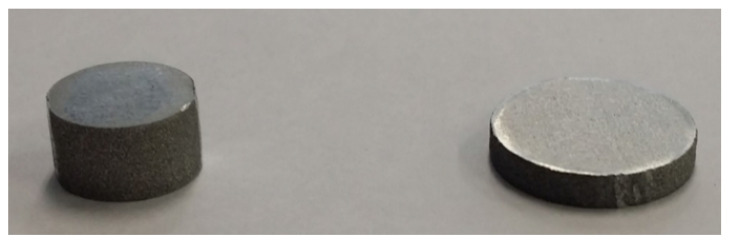
Test specimen used for the SHPB test.

**Figure 25 materials-15-07179-f025:**
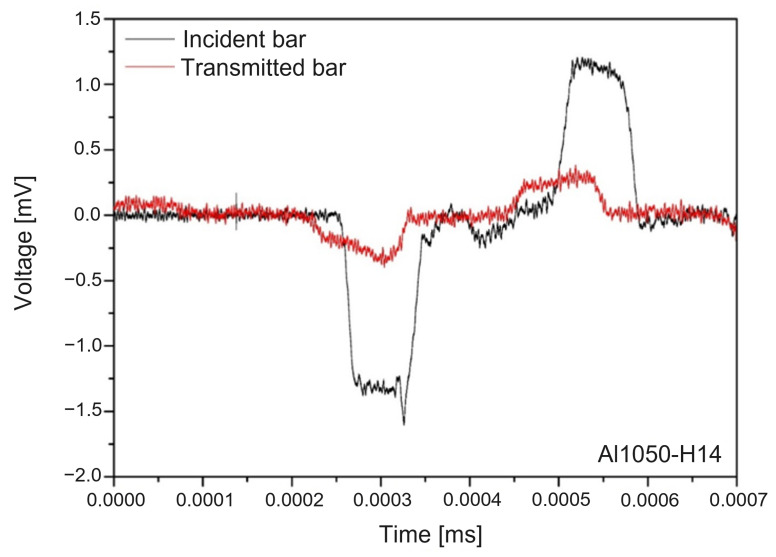
Measured incident and transmitted waves with aluminum 1050-H14.

**Figure 26 materials-15-07179-f026:**
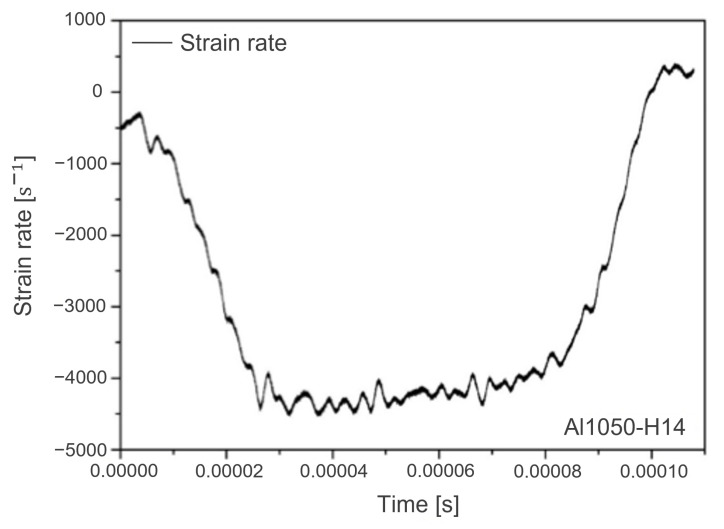
Measured strain rates with aluminum 1050-H14.

**Figure 27 materials-15-07179-f027:**
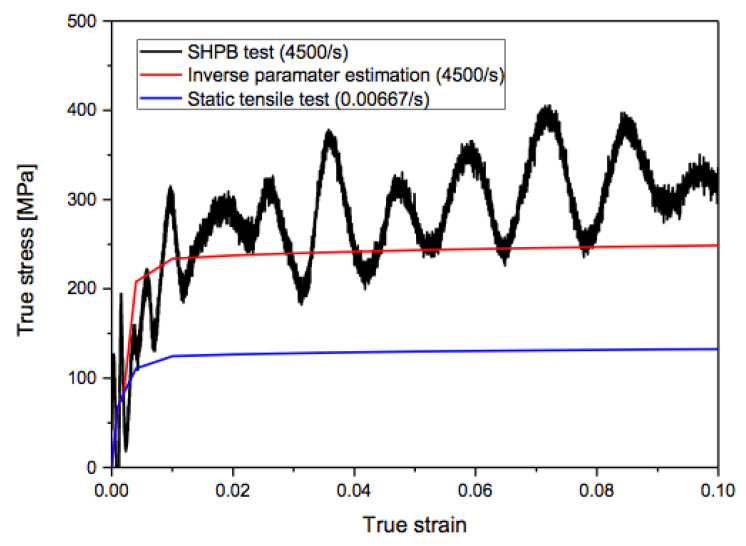
Comparison of experimental and numerical dynamic strain-stress data with aluminum 1050-H14.

**Table 1 materials-15-07179-t001:** Material properties for EMF numerical simulation.

	Property	Value	Unit
Coil (copper)	Resistivity	1.72 × 10^−8^	Ωm
Sheet (A11050-H14)	Resistivity	2.82 × 10^−8^	Ωm
	Possion’s ratio	0.35	
	Density	2980	$kg/m^{3}$
	Elastic modulus	69.0	GPa

**Table 2 materials-15-07179-t002:** Optimal estimates of Kriging model parameters.

	${\hat{\theta }}_{1}$ (*C*)	${\hat{\theta }}_{2}$ (*p*)	${\hat{\sigma }}^{2}$	$\hat{\mu }$
$a_{1}$	1.680248 $\times \;10^{-1}$	6.356147 $\times \;10^{-2}$	5.429284 $\times \;10^{3}$	1.101648 $\times \;10^{0}$
$a_{2}$	2.333454 $\times \;10^{-1}$	8.448063 $\times \;10^{-1}$	6.113093 $\times \;10^{0}$	9.340040 $\times \;10^{-1}$

**Table 3 materials-15-07179-t003:** Numerical verification results of the constructed Kriging models with the testing data.

	$R^{2}$	MARE
$a_{1}$	9.994027 $\times \;10^{-1}$	3.449596 $\times \;10^{-2}$
$a_{2}$	9.687801 $\times \;10^{-1}$	1.473284 $\times \;10^{-1}$

**Table 4 materials-15-07179-t004:** Numerical verification results of the constructed reduced EMF simulation with the training and testing data.

	Training Data		Testing Data	
	$R^{2}$	MARE	$R^{2}$	MARE
Minimum	9.998999 $\times \;10^{-1}$	3.540579 $\times \;10^{-3}$	9.995423 $\times \;10^{-1}$	2.972685 $\times \;10^{-2}$
Maximum	9.999990 $\times \;10^{-1}$	2.277471 $\times \;10^{-1}$	9.999966 $\times \;10^{-1}$	1.641368 $\times \;10^{0}$
Average	9.999802 $\times \;10^{-1}$	1.117795 $\times \;10^{-1}$	9.999375 $\times \;10^{-1}$	1.910738 $\times \;10^{-1}$

**Table 5 materials-15-07179-t005:** Optimal estimates of the modified Johnson–Cook constitutive parameters $\hat{C}$ and $\hat{p}$ along with the optimal regularization parameter α *.

$\hat{C}$	$\hat{p}$	α *
6.6452 $\times \;10^{-3}$	1.6318 $\times \;10^{0}$	3.8367 $\times \;10^{-1}$

## Data Availability

The data presented in this study are available on request from the corresponding author.

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
