# Peer review of "Inverse Identification of a Constitutive Model for High-Speed Forming Simulation: An Application to Electromagnetic Metal Forming"

_materials, 2022, doi:10.3390/ma15207179_

Round 1
Reviewer 1 Report
A very interesting paper dealing with problems associated with numerical simulation of high-speed forming processes such as electromagnetic metal forming (EMF) and electrohydraulic forming (EHF). In order to increase accuracy and shorten computational time authors proposed a new strategy - inverse constitutive modeling in the form of nonlinear least squares with regularization. The presented mathematical model is general and applicable to any high-speed forming process. The methodology and results are well described and based partially on previous work by the authors.
In general, this article is a solid work with nothing much to review. But, I have some questions for the authors:
- How did you measure and control the blank holder force during the experiment?
- Did you take friction into account when modeling the EMF process?
Please capitalize on letter at the begging of sentence - line 258.
Author Response
Dear Reviewer 1,
I appreciate your allowing us to submit a revised manuscript, “Inverse Identification of a Constitutive Model for High-Speed Forming Simulation: an Application to Electromagnetic Metal Forming,” under consideration by Materials.
Thank you for your time and effort in reviewing our manuscript and providing us with valuable comments so that we can improve the manuscript.
We revised the submitted manuscript to reflect all your feedback, and changes relevant to your comments are colored in blue in the revised manuscript. We also altered other parts to improve the draft, and these changes are colored in orange in the revised manuscript.
Please see the attachment for our responses to your comments and questions. We placed the revised manuscript after the responses in a PDF file.

Reviewer 2 Report
The manuscript can be accepted after a minor revision:
1. There are mentioned some specific functions in 5.2: "fmincon", "SQP". An explanation is necessary or a reference
2. A discussion on how grid level influence the solution for EMF simulations is necessary.
3. I recommend to put axis labels in every figure
4. A more carefull review of the actual literature is necessary in order to compare the model presented in this manuscript to other contributions. It seems that the authors made the literature survey just till 2016. There are also more actual contributions in the field. For example: Mahmoud, M.; Bay, F.; Pino Muñoz, D. An Efficient Computational Model for Magnetic Pulse Forming of Thin Structures. Materials 2021, 14, 7645
Author Response
Dear Reviewer 2,
I appreciate your allowing us to submit a revised manuscript, “Inverse Identification of a Constitutive Model for High-Speed Forming Simulation: an Application to Electromagnetic Metal Forming,” under consideration by Materials.
Thank you for your time and effort in reviewing our manuscript and providing us with valuable comments so that we can improve the manuscript.
We revised the submitted manuscript to reflect all your feedback, and changes relevant to your comments are colored in red in the revised manuscript. We also altered other parts to improve the draft, and these changes are colored in orange in the revised manuscript.
Please see the attachment for our responses to your comments and questions. We placed the revised manuscript after the responses in a PDF file.

Reviewer 3 Report
|
In this paper, the authors formulated an inverse parameter estimation problem to identify constitutive parameters in modified Johnson-Cook model used in high-speed forming simulation, they effectively solved the problem by regularized nonlinear least squares with the help of an L-curve and a reduced-order model. The practicability of the identifying constitutive parameters was validated by EMF free bulge test and SHPB test results. The manuscript was well written, and the conclusions were convincing. Some revisions are suggested. |
1. In Section 4.1, the locations of the basis vectors on the sheet should be denoted, and it should be further discussed that why microscopic fluctuating deformation represented by the second basis vector provides dominant influence.
2. In Fig. 13, the titles of the coordinates should be provided.
3. In Line 493, “111.198 and 111” seems should be deleted.
4. In Line 507, “valle” seems should be revised by “valley”.
5. In Eq. (27)-(29), each variable should be defined.
Author Response
Dear Reviewer 3,
I appreciate your allowing us to submit a revised manuscript, “Inverse Identification of a Constitutive Model for High-Speed Forming Simulation: an Application to Electromagnetic Metal Forming,” under consideration by Materials.
Thank you for your time and effort in reviewing our manuscript and providing us with valuable comments so that we can improve the manuscript.
We revised the submitted manuscript to reflect all your feedback, and changes relevant to your comments are colored in violet in the revised manuscript. We also altered other parts to improve the draft, and these changes are colored in orange in the revised manuscript.
Please see the attachment for our responses to your comments. We placed the revised manuscript after the responses in a PDF file.
